# Balancing Suspense and Surprise: Timely Decision Making with Endogenous Information Acquisition

**Ahmed M. Alaa**
Electrical Engineering Department
University of California, Los Angeles

**Mihaela van der Schaar**
Electrical Engineering Department
University of California, Los Angeles

## Abstract

We develop a Bayesian model for decision-making under time pressure with endogenous information acquisition. In our model, the decision-maker decides *when* to observe (costly) information by sampling an underlying continuous-time stochastic process (time series) that conveys information about the potential occurrence/non-occurrence of an adverse event which will terminate the decision-making process. In her attempt to predict the occurrence of the adverse event, the decision-maker follows a policy that determines *when to acquire information* from the time series (continuation), and *when to stop* acquiring information and make a final prediction (stopping). We show that the optimal policy has a "*rendezvous*" structure, i.e. a structure in which whenever a new information sample is gathered from the time series, the optimal "date" for acquiring the next sample becomes computable. The optimal interval between two information samples balances a trade-off between the decision maker's "surprise", i.e. the drift in her posterior belief after observing new information, and "suspense", i.e. the probability that the adverse event occurs in the time interval between two information samples. Moreover, we characterize the continuation and stopping regions in the decision-maker's state-space, and show that they depend not only on the decision-maker's beliefs, but also on the "context", i.e. the current realization of the time series.

## 1   Introduction

The problem of timely risk assessment and decision-making based on a sequentially observed time series is ubiquitous, with applications in finance, medicine, cognitive science and signal processing [1-7]. A common setting that arises in all these domains is that a decision-maker, provided with sequential observations of a time series, needs to decide whether or not an *adverse event* (e.g. financial crisis, clinical acuity for ward patients, etc) will take place in the future. The decision-maker's recognition of a forthcoming adverse event needs to be timely, for that a delayed decision may hinder effective intervention (e.g. delayed admission of clinically acute patients to intensive care units can lead to mortality [5]). In the context of cognitive science, this decision-making task is known as the *two-alternative forced choice* (2AFC) task [15]. Insightful structural solutions for the optimal Bayesian 2AFC decision-making policies have been derived in [9-16], most of which are inspired by the classical work of Wald on sequential probability ratio tests (SPRT) [8].

In this paper, we present a Bayesian decision-making model in which a decision-maker adaptively decides *when* to gather (costly) information from an underlying time series in order to accumulate evidence on the occurrence/non-occurrence of an adverse event. The decision-maker operates under time pressure: occurrence of the adverse event terminates the decision-making process. Our abstract model is motivated and inspired by many practical decision-making tasks such as: constructing temporal patterns for gathering sensory information in perceptual decision-making [1], scheduling lab

tests for ward patients in order to predict clinical deterioration in a timely manner [3, 5], designing breast cancer screening programs for early tumor detection [7], etc.

We characterize the structure of the optimal decision-making policy that prescribes when should the decision-maker acquire new information, and when should she stop acquiring information and issue a final prediction. We show that the decision-maker's posterior belief process, based on which policies are prescribed, is a supermartingale that reflects the decision-maker's tendency to deny the occurrence of an adverse event in the future as she observes the survival of the time series for longer time periods. Moreover, the information acquisition policy has a "*rendezvous*" structure; the optimal "date" for acquiring the next information sample can be computed given the current sample. The optimal schedule for gathering information over time balances the information gain (surprise) obtained from acquiring new samples, and the probability of survival for the underlying stochastic process (suspense). Finally, we characterize the continuation and stopping regions in the decision-maker's state-space and show that, unlike previous models, they depend on the time series "context" and not just the decision-maker's beliefs.

**Related Works** Mathematical models and analyses for perceptual decision-making based on sequential hypothesis testing have been developed in [9-17]. Most of these models use tools from sequential analysis developed by Wald [8] and Shiryaev [21, 22]. In [9,13,14], optimal decision-making policies for the 2AFC task were computed by modelling the decision-maker's sensory evidence using diffusion processes [20]. These models assume an infinite time horizon for the decision-making policy, and an exogenous supply of sensory information.

The assumption of an infinite time horizon was relaxed in [10] and [15], where decision-making is assumed to be performed under the pressure of a stochastic deadline; however, these deadlines were considered to be drawn from known distributions that are independent of the hypothesis and the realized sensory evidence, and the assumption of an exogenous information supply was maintained. In practical settings, the deadlines would naturally be dependent on the realized sensory information (e.g. patients' acuity events are correlated with their physiological information [5]), which induces more complex dynamics in the decision-making process. Context-based decision-making models were introduced in [17], but assuming an exogenous information supply and an infinite time horizon.

The notions of "suspense" and "surprise" in Bayesian decision-making have also been recently introduced in the economics literature (see [18] and the references therein). These models use measures for *Bayesian surprise*, originally introduced in the context of sensory neuroscience [19], in order to model the explicit preference of a decision-maker to non-instrumental information. The goal there is to design information disclosure policies that are suspense-optimal or surprise-optimal. Unlike our model, such models impose suspense (and/or surprise) as a (behavioral) preference of the decision-maker, and hence they do not emerge endogenously by virtue of rational decision making.

## 2 Timely Decision Making with Endogenous Information Gathering

**Time Series Model** The decision-maker has access to a time-series $X(t)$ modeled as a continuous-time stochastic process that takes values in $\mathbb{R}$, and is defined over the time domain $t \in \mathbb{R}_+$, with an underlying filtered probability space $(\Omega, \mathcal{F}, \{\mathcal{F}_t\}_{t \in \mathbb{R}_+}, \mathbb{P})$. The process $X(t)$ is naturally adapted to $\{\mathcal{F}_t\}_{t \in \mathbb{R}_+}$, and hence the filtration $\mathcal{F}_t$ abstracts the information conveyed in the time series realization up to time $t$. The decision-maker extracts information from $X(t)$ to guide her actions over time.

We assume that $X(t)$ is a stationary Markov process[1], with a stationary transition kernel $\mathbb{P}_\theta (X(t) \in A | \mathcal{F}_s) = \mathbb{P}_\theta (X(t) \in A | X(s)), \forall A \subset \mathbb{R}, \forall s < t \in \mathbb{R}_+$, where $\theta$ is a realization of a latent Bernoulli random variable $\Theta \in \{0, 1\}$ (unobservable by the decision-maker), with $\mathbb{P}(\Theta = 1) = p$. The distributional properties of the paths of $X(t)$ are determined by $\theta$, since the realization of $\theta$ decides which Markov kernel ($\mathbb{P}_o$ or $\mathbb{P}_1$) generates $X(t)$. If the realization $\theta$ is equal to 1, then an adverse event occurs almost surely at a (finite) random time $\tau$, the distribution of which is dependent on the realization of the path $(X(t))_{0 \le t \le \tau}$.

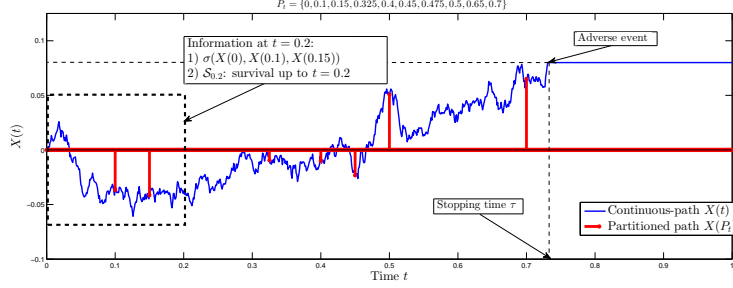

Figure 1: An exemplary stopped sample path for $X^\tau(t)|\Theta = 1$, with an exemplary partition $P_t$.

The decision-maker's ultimate goal is to sequentially observe $X(t)$, and infer $\theta$ before the adverse event happens; inference is obsolete if it is declared after $\tau$. Since $\Theta$ is latent, the decision-maker is unaware whether the adverse event will occur or not, i.e. whether her access to $X(t)$ is temporary ($\tau < \infty$ for $\theta = 1$) or permanent ($\tau = \infty$ for $\theta = 0$). In order to model the occurrence of the adverse event; we define $\tau$ as an $\mathcal{F}$-stopping time for the process $X(t)$, for which we assume the following:

- The stopping time $\tau|\Theta = 1$ is finite almost surely, whereas $\tau|\Theta = 0$ is infinite almost surely, i.e. $\mathbb{P}(\tau < \infty|\Theta = 1) = 1$, and $\mathbb{P}(\tau = \infty|\Theta = 0) = 1$.

- The stopping time $\tau|\Theta = 1$ is *accessible*[2], with a Markovian dependency on history, i.e. $\mathbb{P}(\tau < t|\mathcal{F}_s) = \mathbb{P}(\tau < t|X(s))$, $\forall s < t$, where $\mathbb{P}(\tau < t|X(s))$ is an injective map from $\mathbb{R}$ to $[0, 1]$ and $\mathbb{P}(\tau < t|X(s))$ is non-decreasing in $X(s)$.

Thus, unlike the stochastic deadline models in [10] and [15], the decision deadline in our model (i.e. occurrence of the adverse event) is *context-dependent* as it depends on the time series realization (i.e. $\mathbb{P}(\tau < t|X(s))$ is not independent of $X(t)$ as in [15]). We use the notation $X^\tau(t) = X(t \wedge \tau)$, where $t \wedge \tau = \min\{t, \tau\}$ to denote the stopped process to which the decision-maker has access. Throughout the paper, the measures $\mathbb{P}_o$ and $\mathbb{P}_1$ assign probability measures to the paths $X^\tau(t)|\Theta = 0$ and $X^\tau(t)|\Theta = 1$ respectively, and we assume that $\mathbb{P}_o << \mathbb{P}_1$[3].

**Information** The decision-maker can only observe a set of (costly) samples of $X^\tau(t)$ rather than the full continuous path. The samples observed by the decision-maker are captured by partitioning $X(t)$ over specific time intervals: we define $P_t = \{t_o, t_1, \ldots, t_{N(P_t)-1}\}$, with $0 \le t_o < t_1 < \ldots < t_{N(P_t)-1} \le t$, as a size-$N(P_t)$ partition of $X^\tau(t)$ over the interval $[0, t]$, where $N(P_t)$ is the total number of samples in the partition $P_t$. The decision-maker observes the values that $X^\tau(t)$ takes at the time instances in $P_t$; thus the sequence of observations is given by the process $X(P_t) = \sum_{i=0}^{N(P_t)-1} X(t_i)\delta_{t_i}$, where $\delta_{t_i}$ is the Dirac measure. The space of all partitions over the interval $[0, t]$ is denoted by $\mathcal{P}_t = [0, t]^{\mathbb{N}}$. We denote the probability measures for partitioned paths generated under $\Theta = 0$ and 1 with a partition $P_t$ as $\tilde{\mathbb{P}}_o(P_t)$ and $\tilde{\mathbb{P}}_1(P_t)$ respectively.

Since the decision-maker observes $X^\tau(t)$ through the partition $P_t$, her information at time $t$ is conveyed in the $\sigma$-algebra $\sigma(X^\tau(P_t)) \subset \mathcal{F}_t$. The stopping event is observable by the decision-maker even if $\tau \notin P_\tau$. We denote the $\sigma$-algebra generated by the stopping event as $\mathcal{S}_t = \sigma(\mathbf{1}_{\{t \ge \tau\}})$. Thus, the information that the decision-maker has at time $t$ is expressed by the filtration $\tilde{\mathcal{F}}_t = \sigma(X^\tau(P_t)) \vee \mathcal{S}_t$. Hence, any decision-making policy needs to be $\tilde{\mathcal{F}}_t$-measurable.

Figure 1 depicts a Brownian path (a sample path of a Wiener process, which satisfies all the assumptions of our model)[4], with an exemplary partition $P_t$ over the time interval $[0, 1]$. The decision-maker observes the samples in $X(P_t)$ sequentially, and reasons about the realization of the latent variable $\Theta$ based on these samples and the process survival, i.e. at $t = 0.2$, the decision-maker's information resides in the $\sigma$-algebra $\sigma(X(0), X(0.1), X(0.15))$ generated by the samples

in $P_{0.2} = \{0, 0.1, 0.15\}$, and the $\sigma$-algebra generated by the process' survival $\mathcal{S}_{0.2} = \sigma(\mathbf{1}_{\{\tau > 0.2\}})$.

**Policies and Risks** The decision-maker's goal is to come up with a (timely) decision $\hat{\theta} \in \{0, 1\}$, that reflects her prediction for whether the actual realization $\theta$ is 0 or 1, before the process $X^\tau(t)$ potentially stops at the unknown time $\tau$. The decision-maker follows a *policy*: a (continuous-time) mapping from the observations gathered up to every time instance $t$ to two types of actions:

- A *sensing action* $\delta_t \in \{0, 1\}$: if $\delta_t = 1$, then the decision-maker decides to observe a new sample from the running process $X^\tau(t)$ at time $t$.
- A *continuation/stopping action* $\hat{\theta}_t \in \{\emptyset, 0, 1\}$: if $\hat{\theta}_t \in \{0, 1\}$, then the decision-maker decides to *stop* gathering samples from $X^\tau(t)$, and declares a final decision (estimate) for $\theta$. Whenever $\hat{\theta}_t = \emptyset$, the decision-maker *continues* observing $X^\tau(t)$ and postpones her declaration for the estimate of $\theta$.

A policy $\pi = (\pi_t)_{t \in \mathbb{R}_+}$ is a ($\tilde{\mathcal{F}}_t$-measurable) mapping rule that maps the information in $\tilde{\mathcal{F}}_t$ to an action tuple $\pi^t = (\delta_t, \hat{\theta}_t)$ at every time instance $t$. We assume that every single observation that the decision-maker draws from $X^\tau(t)$ entails a fixed cost, hence the process $(\delta_t)_{t \in \mathbb{R}_+}$ has to be a point process under any optimal policy[5]. We denote the space of all such policies by $\Pi$.

A policy $\pi$ generates the following random quantities as a function of the paths $X^\tau(t)$ on the probability space $(\Omega, \mathcal{F}, \{\mathcal{F}_t\}_{t \in \mathbb{R}_+}, \mathbb{P})$:

**1- A stopping time $T_\pi$:** The first time at which the decision-maker declares its estimate for $\theta$, i.e. $T_\pi = \inf\{t \in \mathbb{R}_+ : \hat{\theta}_t \in \{0, 1\}\}$.
**2- A decision (estimate of $\theta$) $\hat{\theta}_\pi$:** Given by $\hat{\theta}_\pi = \hat{\theta}_{T_\pi \wedge \tau}$.
**3- A random partition $P_{T_\pi}^\pi$:** A realization of the point process $(\delta_t)_{t \in \mathbb{R}_+}$, comprising a finite set of strictly increasing $\mathcal{F}$-stopping times at which the decision-maker decides to sample the path $X^\tau(t)$.

A loss function is associated with every realization of the policy $\pi$, representing the overall cost incurred when following that policy for a specific path $X^\tau(t)$. The loss function is given by

$$\ell(\pi; \Theta) \triangleq (C_1 \underbrace{\mathbf{1}_{\{\hat{\theta}_\pi = 0, \theta = 1\}}}_{\text{Type I error}} + C_o \underbrace{\mathbf{1}_{\{\hat{\theta}_\pi = 1, \theta = 0\}}}_{\text{Type II error}} + \underbrace{C_d T_\pi}_{\text{Delay}}) \mathbf{1}_{\{T_\pi \leq \tau\}} + \underbrace{C_r \mathbf{1}_{\{T_\pi > \tau\}}}_{\text{Deadline missed}} + \underbrace{C_s N(P_{T_\pi \wedge \tau}^\pi)}_{\text{Information}},$$

(1)

where $C_1$ is the cost of type I error (failure to anticipate the adverse event), $C_o$ is the cost of type II error (falsely predicting that an adverse event will occur), $C_d$ is the cost of the delay in declaring the estimate $\hat{\theta}_\pi$, $C_r$ is the cost incurred when the adverse event occurs before an estimate $\hat{\theta}_\pi$ is declared (cost of missing the deadline), and $C_s$ is the cost of every observation sample (cost of information). The risk of each policy $\pi$ is defined as its expected loss

$$R(\pi) \triangleq \mathbb{E}\left[\ell(\pi; \Theta)\right],$$

(2)

where the expectation is taken over the paths of $X^\tau(t)$. In the next section, we characterize the structure of the optimal policy $\pi^* = \arg\inf_{\pi \in \Pi} R(\pi)$.

## 3 Structure of the Optimal Policy

Since the decision-maker's posterior belief at time $t$, defined as $\mu_t = \mathbb{P}(\Theta = 1 | \tilde{\mathcal{F}}_t)$, is an important statistic for designing sequential policies [10, 21-22], we start our characterization for $\pi^*$ by investigating the belief process $(\mu_t)_{t \in \mathbb{R}_+}$.

### 3.1 The Posterior Belief Process

Recall that the decision-maker distills information from two types of observations: the realization of the partitioned time series $X^\tau(P_t)$ (i.e. the information in $\sigma(X^\tau(P_t))$), and 2) the survival of the

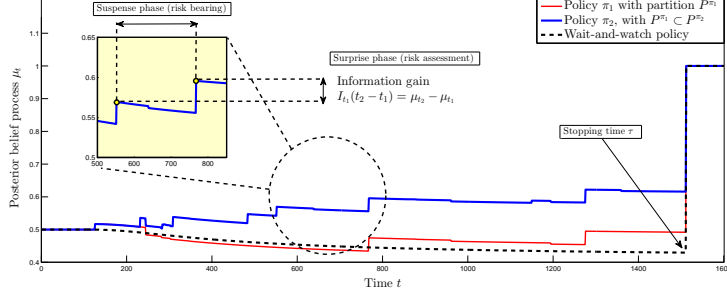

Figure 2: Depiction for exemplary belief paths of different policies under $\Theta = 1$.

process up to time $t$ (i.e. the information in $\mathcal{S}_t$). In the following Theorem, we study the evolution of the decision-maker's beliefs as she integrates these pieces of information over time[6].

**Theorem 1 (Information and beliefs).** Every posterior belief trajectory $(\mu_t)_{t \in \mathbb{R}_+}$ associated with a policy $\pi \in \Pi$ that creates a partition $P_t^\pi \in \mathcal{P}_t$ of $X^\tau(t)$ is a càdlàg path given by

$$
\mu_t = \mathbf{1}_{\{t \geq \tau\}} + \mathbf{1}_{\{0 \leq t < \tau\}} \cdot \left( 1 + \frac{1-p}{p} \cdot \frac{d\tilde{\mathbb{P}}_o(P_t^\pi)}{d\tilde{\mathbb{P}}_1(P_t^\pi)} \right)^{-1},
$$

where $\frac{d\tilde{\mathbb{P}}_o(P_t^\pi)}{d\tilde{\mathbb{P}}_1(P_t^\pi)}$ is the Radon–Nikodym derivative[7] of the measure $\tilde{\mathbb{P}}_o(P_t^\pi)$ with respect to $\tilde{\mathbb{P}}_1(P_t^\pi)$, and is given by the following elementary predictable process

$$
\frac{1}{\frac{d\tilde{\mathbb{P}}_o(P_t^\pi)}{d\tilde{\mathbb{P}}_1(P_t^\pi)}} = \sum_{k=1}^{N(P_t^\pi)-1} \underbrace{\frac{\mathbb{P}(X(P_t^\pi)|\Theta=1)}{\mathbb{P}(X(P_t^\pi)|\Theta=0)}}_{\text{Likelihood ratio}} \underbrace{\mathbb{P}(\tau > t | \sigma(X(P_t^\pi), \Theta=1)}_{\text{Survival probability}} \mathbf{1}_{\{P_t^\pi(k) \leq t \leq P_t^\pi(k+1)\}},
$$

for $t \geq P_t^\pi(1)$, and $p\,\mathbb{P}(\tau > t | \Theta = 1)$ for $t < P_t^\pi(k)$. Moreover, the path $(\mu_t)_{t \in \mathbb{R}_+}$ has exactly $N(P_{T_\pi \wedge \tau}^\pi) + \mathbf{1}_{\{\tau < \infty\}}$ jumps at the time indexes in $P_{t \wedge \tau}^\pi \cup \{\tau\}$. □

Theorem 1 says that every belief path is right-continuous with left limits, and has jumps at the time indexes in the partition $P_t^\pi$, whereas between each two jumps, the paths $(\mu_t)_{t \in [t_1, t_2)}, t_1, t_2 \in P_t^\pi$ are predictable (i.e. they are known ahead of time once we know the magnitudes of the jumps preceding them). This means that the decision-maker obtains "active" information by probing the time series to observe new samples (i.e. the information in $\sigma(X^\tau(P_t))$), inducing jumps that revive her beliefs, whereas the progression of time without witnessing a stopping event offers the decision-maker "passive information" that is distilled just from the costless observation of process survival information. Both sources of information manifest themselves in terms of the likelihood ratio, and the survival probability in the expression of $\frac{d\mathbb{P}_o(P_t^\pi)}{d\mathbb{P}_1(P_t^\pi)}$ above.

In Figure 2, we plot the càdlàg belief paths for policies $\pi_1$ and $\pi_2$, where $P^{\pi_1} \subset P^{\pi_2}$ (i.e. policy $\pi_1$ observe a subset of the samples observed by $\pi_2$). We also plot the (predictable) belief path of a *wait-and-watch* policy that observes no samples. We can see that $\pi_2$, which has more jumps of "active information", copes faster with the truthful belief over time. Between each two jumps, the belief process exhibits a non-increasing predictable path until fed with a new piece of information. The wait-and-watch policy has its belief drifting away from the prior $p = 0.5$ towards the wrong belief $\mu_t = 0$ since it only distills information from the process survival, which favors the hypothesis $\Theta = 0$. This discussion motivates the introduction of the following key quantities.

**Information gain (surprise) $I_t(\Delta t)$:** The amount of drift in the decision-maker's belief at time $t + \Delta t$ with respect to her belief at time $t$, given the information available up to time $t$, i.e. $I_t(\Delta t) = (\mu_{t+\Delta t} - \mu_t)|\tilde{\mathcal{F}}_t$.

**Posterior survival function (suspense)** $S_t(\Delta t)$**:** The probability that a process generated with $\Theta = 1$ survives up to time $t + \Delta t$ given the information observed up to time $t$, i.e. $S_t(\Delta t) = \mathbb{P}(\tau > t + \Delta t | \tilde{\mathcal{F}}_t, \Theta = 1)$. The function $S_t(\Delta t)$ is a non-increasing function in $\Delta t$, i.e. $\frac{\partial S_t(\Delta t)}{\partial \Delta t} \leq 0$.

That is, the information gain is the amount of "surprise" that the decision-maker experiences in response to a new information sample expressed in terms of the change in here belief, i.e. the jumps in $\mu_t$, whereas the survival probability (suspense) is her assessment for the risk of having the adverse event taking places in the next $\Delta t$ time interval. As we will see in the next subsection, the optimal policy would balance the two quantities when scheduling the times to sense $X^\tau(t)$.

We conclude our analysis for the process $\mu_t$ by noting that lack of information samples creates bias towards the belief that $\Theta = 0$ (e.g. see the belief path of the wait-and-watch policy in Figure 2). We formally express this behavior in the following Corollary.

**Corollary 1 (Leaning towards denial).** For every policy $\pi \in \Pi$, the posterior belief process $\mu_t$ is a *supermartingale* with respect to $\tilde{\mathcal{F}}_t$, where

$$\mathbb{E}[\mu_{t+\Delta t} | \tilde{\mathcal{F}}_t] = \mu_t - \mu_t^2 S_t(\Delta t)(1 - S_t(\Delta t)) \leq \mu_t, \ \forall \Delta t \in \mathbb{R}_+. \qquad \square$$

Thus, unlike classical Bayesian learning models with a belief martingale [18, 21-23], the belief process in our model is a supermartingale that leans toward decreasing over time. The reason for this is that in our model, time conveys information. That is, unlike [10] and [15] where the decision deadline is hypothesis-independent and is almost surely occurring in finite time for any path, in our model the occurrence of the adverse event is itself a hypothesis, hence observing the survival of the process is informative and contributes to the evolution of the belief. The informativeness of both the acquired information samples and process survival can be disentangled using Doob decomposition, by writing $\mu_t$ as $\mu_t = \tilde{\mu}_t + A(\mu_t, S_t(\Delta t))$, where $\tilde{\mu}_t$ is a martingale, capturing the information gain from the acquired samples, and $A(\mu_t, S_t(\Delta t))$ is a predictable *compensator* process [23], capturing information extracted from the process survival.

## 3.2   The Optimal Policy

The optimal policy $\pi^*$ minimizes the expected risk as defined in (1) and (2) by generating the tuple of random processes $(T_\pi, \hat{\theta}_\pi, P_t^\pi)$ in response to the paths of $X^\tau(t)$ on $(\Omega, \mathcal{F}, \{\mathcal{F}_t\}_{t \in \mathbb{R}_+}, \mathbb{P})$ in a way that "shapes" a belief process $\mu_t$ that maximizes informativeness, maintains timeliness and controls cost. In the following, we introduce the notion of a "rendezvous policy", then in Theorem 2, we show that the optimal policy $\pi^*$ complies with this definition.

**Rendezvous policies** We say that a policy $\pi$ is a *rendezvous* policy, if the random partition $P_{T_\pi}^\pi$ constructed by the sequence of sensing actions $(\delta_t^\pi)_{t \in [0, T_\pi]}$, is a point process with predictable jumps, where for every two consecutive jumps at times $t$ and $t'$, with $t' > t$ and $t, t' \in P_{T_\pi}^\pi$, we have that $t'$ is $\tilde{\mathcal{F}}_t$-measurable.

That is, a rendezvous policy is a policy that constructs a sensing schedule $(\delta_t^\pi)_{t \in [0, T_\pi]}$, such that every time $t'$ at which the decision-maker acquires information is actually computable using the information available up to time $t$, the previous time instance at which information was gathered. Hence, the decision-maker can decide the next "date" in which she will gather information directly after she senses a new information sample. This structure is a natural consequence of the information structure in Theorem 1, since the belief paths between every two jumps are predictable, then they convey no "actionable" information, i.e. if the decision-maker was to respond to a predictable belief path, say by sensing or making a stopping decision, then she should have taken that decision right before the predictable path starts, which leads her to better off by saving the delay cost $C_d$. We denote the space of all rendezvous policies by $\Pi_r$. In the following Theorem, we establish that the rendezvous structure is optimal.

**Theorem 2 (Rendezvous).** The optimal policy $\pi^*$ is a rendezvous policy ($\pi^* \in \Pi_r$). $\qquad \square$

A direct implication of Theorem 2 is that the time variable can now be viewed as a state variable, whereas the problem is virtually solved in "discrete-time" since the decision-maker effectively jumps from one time instance to another in a discrete manner. Hence, we alter the definition of the action $\delta_t$ from an indicator variable that indicates sensing the time series at time $t$, to a "rendezvous action" that takes real values, and specifies the time after which the decision-maker would sense a new sample, i.e. if $\delta_t = \Delta t$, then the decision-maker gathers the new sample at $t + \Delta t$. This transformation restricts our policy design problem to the space of rendezvous policies $\Pi_r$, which we know from Theorem 2 that it contains the optimal policy (i.e. $\pi^* = \arg\inf_{\pi \in \Pi_r} R(\pi)$).

Having established the result in Theorem 2, in the following Theorem, we characterize the optimal policy $\pi^*$ in terms of the random process $(T_{\pi^*}, \hat{\theta}_{\pi^*}, P_t^{\pi^*})$ using discrete-time Bellman optimality conditions [24].

**Theorem 3 (The optimal policy).** The optimal policy $\pi^*$ is a sequence of actions $(\hat{\theta}_t^{\pi^*}, \delta_t^{\pi^*})_{t \in \mathbb{R}_+}$, resulting in a random process $(\hat{\theta}_{\pi^*}, T_{\pi^*}, P_{T_{\pi^*}}^{\pi^*})$ with the following properties:

*(Continuation and stopping)*

1. The process $(t, \mu_t, \bar{X}(P_t^{\pi^*}))_{t \in \mathbb{R}_+}$ is a Markov sufficient statistic for the distribution of $(\hat{\theta}_{\pi^*}, T_{\pi^*}, P_{T_{\pi^*}}^{\pi^*})$, where $\bar{X}(P_t^{\pi^*})$ is the most recent sample in the partition $P_t^{\pi^*}$, i.e. $\bar{X}(P_t^{\pi^*}) = X(t^*), t^* = \max P_t^{\pi^*}$.

2. The policy $\pi^*$ recommends *continuation*, i.e. $\hat{\theta}_t^{\pi^*} = \emptyset$, as long as the belief $\mu_t \in \mathcal{C}(t, \bar{X}(P_t^{\pi^*}))$, where $\mathcal{C}(t, \bar{X}(P_t^{\pi^*}))$, is a time and context-dependent *continuation set* with the following properties: $\mathcal{C}(t', X) \subset \mathcal{C}(t, X), \forall t' > t$, and $\mathcal{C}(t, X') \subset \mathcal{C}(t, X), \forall X' > X$.

*(Rendezvous and decisions)*

1. Whenever $\mu_t \in \mathcal{C}(t, \bar{X}(P_t^{\pi^*}))$, and $t \in P_{T_{\pi^*}}^{\pi^*}$, then the rendezvous $\delta_t^{\pi^*}$ is set as follows

$$\delta_t^{\pi^*} = \arg\inf_{\delta \in \mathbb{R}_+} f(\mathbb{E}[I_t(\delta)], S_t(\delta)),$$

where $f(\mathbb{E}[I_t(\delta)], S_t(\delta))$ is decreasing in $\mathbb{E}[I_t(\delta)]$ and $S_t(\delta)$.

2. Whenever $\mu_t \notin \mathcal{C}(t, \bar{X}(P_t^{\pi^*}))$, then a decision $\hat{\theta}_t^{\pi^*} = \hat{\theta}_{\pi^*} \in \{0, 1\}$ is issued, and is based on a belief threshold as follows: $\hat{\theta}_{\pi^*} = \mathbf{1}_{\left\{ \mu_t \geq \frac{C_1}{C_o + C_1} \right\}}$. The stopping time is given by

$$T_{\pi^*} = \inf\{t \in \mathbb{R}_+ : \mu_t \notin \mathcal{C}(t, \bar{X}(P_t^{\pi^*}))\}. \qquad \square$$

Theorem 3 establishes the structure of the optimal policy and its prescribed actions in the decision-maker's state-space. The first part of the Theorem says that in order to generate the random tuple $(T_{\pi^*}, \hat{\theta}_{\pi^*}, P_t^{\pi^*})$ optimally, we only need to keep track of the realization of the process $(t, \mu_t, \bar{X}(P_t))_{t \in \mathbb{R}_+}$ in every time instance. That is, an optimal policy maps the current belief, the current time, and the most recently observed realization of the time series to an action tuple $(\hat{\theta}_t^\pi, \delta_t^\pi)$, i.e. a decision on whether to stop and declare an estimate for $\theta$ or sense a new sample. Hence, the process $(t, \mu_t, \bar{X}(P_t))_{t \in \mathbb{R}_+}$ represents the "state" of the decision-maker, and the decision-maker's actions can partially influence the state through the belief process, i.e. a decision on when to acquire the next sample affects the distributional properties of the posterior belief. The remaining state variables $t$ and $X(t)$ are beyond the decision-maker's control.

We note that unlike the previous models in [9-16], with the exception of [17], a policy in our model is *context-dependent*. That is, since the state is $(t, \mu_t, \bar{X}(P_t^\pi))$ and not just the time-belief tuple $(t, \mu_t)$, a policy $\pi$ can recommend different actions for the same belief and at the same time but for a different context. This is because, while $\mu_t$ captures what the decision-maker learned from the history, $\bar{X}(P_t^\pi)$ captures her foresightedness into the future, i.e. it can be that the belief $\mu_t$ is not decisive (e.g. $\mu_t \approx p$), but the context is "risky" (i.e. $\bar{X}(P_t^\pi)$ is large), which means that a potential forthcoming adverse event is likely to happen in the near future, hence the decision-maker would be more eager to make a stopping decision and declare an estimate $\hat{\theta}_\pi$. This is manifested through the dependence of the continuation set $\mathcal{C}(t, \bar{X}(P_t^\pi))$ on both time and context; the continuation set is monotonically decreasing in time due to the deadline pressure, and is also monotonically decreasing in $\bar{X}(P_t^\pi)$ due to the dependence of the deadline on the time series realization.

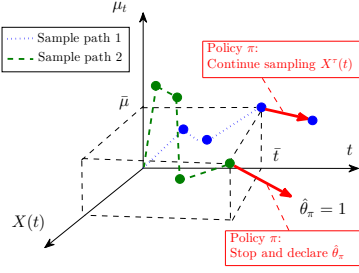

Figure 3: Context-dependence of the policy $\pi$.

The context dependence of the optimal policy is pictorially depicted in Figure 3 where we show two exemplary trajectories for the decision-maker's state, and the actions recommended by a policy $\pi$ for the same time and belief, but a different context, i.e. a stopping action recommended when $X(t)$ is large since it corresponds to a low survival probability, whereas for the same belief and time, a continuation action can be recommended if $X(t)$ is low since it is safer to keep observing the process for that the survival probability is high. Such a prescription specifies optimal decision-making in context-driven settings such as clinical decision-making in critical care environment [3-5], where a combination of a patient's length of hospital stay (i.e. $t$), clinical risk score (i.e. $\mu_t$) and current physiological test measurements (i.e. $\bar{X}(P_t^\pi)$) determine the decision on whether or not a patient should be admitted to an intensive care unit.

The second part of Theorem 3 says that whenever the optimal policy decides to stop gathering information and issue a conclusive decision, it imposes a threshold on the posterior belief, based on which it issues the estimate $\hat{\theta}_{\pi^*}$; the threshold is $\frac{C_1}{C_o + C_1}$, and hence weights the estimates by their respective risks. When the policy favors continuation, it issues a rendezvous action, i.e. the next time instance at which information will be gathered. This rendezvous balances surprise and suspense: the decision-maker prefers maximizing surprise in order to draw the maximum informativeness from the costly sample it will acquire; this is captured in terms of the expected information gain $\mathbb{E}[I_t(\delta)]$. Maximizing surprise may increase suspense, i.e. the probability of process termination, which is controlled by the survival function $S_t(\delta)$, and hence it can be that harvesting the maximum informativeness entails a survival risk when $C_r$ is high. Therefore, the optimal policy selects a rendezvous $\delta_t^{\pi^*}$ that optimizes a combination of the survival risk survival, captured by the cost $C_r$ and the survival function $S_t(\Delta t)$, and the value of information, captured by the costs $C_o$, $C_1$ and the expected information gain $\mathbb{E}[I_t(\delta)]$.

## 4 Conclusions

We developed a model for decision-making with endogenous information acquisition under time pressure, where a decision-maker needs to issue a conclusive decision before an adverse event (potentially) takes place. We have shown that the optimal policy has a "rendezvous" structure, i.e. the optimal policy sets a "date" for gathering a new sample whenever the current information sample is observed. The optimal policy selects the time between two information samples such that it balances the information gain (surprise) with the survival probability (suspense). Moreover, we characterized the optimal policy's continuation and stopping conditions, and showed that they depend on the context and not just on beliefs. Our model can help understanding the nature of optimal decision-making in settings where timely risk assessment and information gathering is essential.

## 5 Acknowledgments

This work was supported by the ONR and the NSF (Grant number: ECCS 1462245).

## Footnotes

[1]Most of the insights distilled from our results would hold for more general dependency structures. However, we keep this assumption to simplify the exposition and maintain the tractability and interpretability of the results.

[2]Our analyses hold if the stopping time is totally inaccessible.

[3]The absolute continuity of $\mathbb{P}_o$ with respect to $\mathbb{P}_1$ means that no sample path of $X^\tau(t)|\Theta = 0$ should be fully revealing of the realization of $\Theta$.

[4]In Figure 1, the stopping event was simulated as a totally inaccessible first jump of a Poisson process.

[5]Note that the cost of observing any local continuous path is infinite, hence any optimal policy must have $(\delta_t)_{t \in \mathbb{R}_+}$ being a point process to keep the number of observed samples finite.

[6] All proofs are provided in the supplementary material

[7] Since we impose the condition $\mathbb{P}_o << \mathbb{P}_1$ and fix a partition $P_t$, then the Radon–Nikodym derivative exists.

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
