[Supplementary Material · Supplementary_NIPS.pdf]

# Balancing Suspense and Surprise: Timely Decision Making with Endogenous Information Acquisition

**Ahmed M. Alaa**
Electrical Engineering Department
University of California, Los Angeles
ahmedmalaa@ucla.edu

**Mihaela van der Schaar**
Electrical Engineering Department
University of California, Los Angeles
mihaela@ee.ucla.edu

## Proofs

### Proof of Theorem 1

The posterior belief process $(\mu_t)_{t \in \mathbb{R}_+}$ is given by

$$
\begin{aligned}
\mu_t &= \mathbb{P}(\Theta = 1 | \tilde{\mathcal{F}}_t) \\
&\overset{(a)}{=} \mathbb{P}(\Theta = 1 | \sigma(X(P_t^\pi)), \mathcal{S}_t) \\
&= \mathbf{1}_{\{t \geq \tau\}} \cdot \mathbb{P}(\Theta = 1 | \sigma(X(P_t^\pi)), t \geq \tau) + \mathbf{1}_{\{t < \tau\}} \cdot \mathbb{P}(\Theta = 1 | \sigma(X(P_t^\pi)), t < \tau) \\
&\overset{(b)}{=} \mathbf{1}_{\{t \geq \tau\}} + \mathbf{1}_{\{t < \tau\}} \cdot \mathbb{P}(\Theta = 1 | \sigma(X(P_t^\pi)), t < \tau),
\end{aligned}
\tag{1}
$$

where we have used the fact that $\tilde{\mathcal{F}}_t = \sigma(X(P_t^\pi)) \vee \mathcal{S}_t$ in (a), and the fact that the event $\{t \geq \tau\}$ is $\tilde{\mathcal{F}}_t$-measurable in (b), and hence $\mathbb{P}(\Theta = 1 | \sigma(X(P_t^\pi)), t \geq \tau) = 1$. Therefore, we can write the posterior belief process $(\mu_t)_{t \in \mathbb{R}_+}$ in the following form

$$
\mu_t = \begin{cases}
1, & \text{for } t \geq \tau \\
\mathbb{P}(\Theta = 1 | \sigma(X(P_t^\pi)), t < \tau), & \text{for } 0 \leq t < \tau.
\end{cases}
$$

Now we focus on computing $\mathbb{P}(\Theta = 1 | \sigma(X(P_t^\pi)), t < \tau)$. Note that using Bayes' rule, we have that

$$
\begin{aligned}
\mathbb{P}(\Theta = 1 | \sigma(X(P_t^\pi)), t < \tau) &= \frac{\mathbb{P}(\Theta = 1, \sigma(X(P_t^\pi)), t < \tau)}{\mathbb{P}(\sigma(X(P_t^\pi)), t < \tau)} \\
&= \frac{\mathbb{P}(\Theta = 1, \sigma(X(P_t^\pi)), t < \tau)}{\sum_{\theta \in \{0,1\}} \mathbb{P}(\Theta = \theta, \sigma(X(P_t^\pi)), t < \tau)} \\
&= \frac{d\mathbb{P}(\sigma(X(P_t^\pi)), t < \tau | \Theta = 1) \, \mathbb{P}(\Theta = 1)}{\sum_{\theta \in \{0,1\}} d\mathbb{P}(\sigma(X(P_t^\pi)), t < \tau | \Theta = \theta) \, \mathbb{P}(\Theta = \theta)} \\
&= \frac{d\mathbb{P}(\sigma(X(P_t^\pi)), t < \tau | \Theta = 1) \, \mathbb{P}(\Theta = 1)}{d\mathbb{P}(\sigma(X(P_t^\pi)), t < \tau | \Theta = 0) \, \mathbb{P}(\Theta = 0) + d\mathbb{P}(\sigma(X(P_t^\pi)), t < \tau | \Theta = 1) \, \mathbb{P}(\Theta = 1)} \\
&= \frac{p \, d\mathbb{P}(\sigma(X(P_t^\pi)), t < \tau | \Theta = 1)}{(1 - p) \, d\mathbb{P}(\sigma(X(P_t^\pi)), t < \tau | \Theta = 0) + p \, d\mathbb{P}(\sigma(X(P_t^\pi)), t < \tau | \Theta = 1)} \\
&= \left(1 + \frac{1 - p}{p} \cdot \frac{d\mathbb{P}(\sigma(X(P_t^\pi)), t < \tau | \Theta = 0)}{d\mathbb{P}(\sigma(X(P_t^\pi)), t < \tau | \Theta = 1)}\right)^{-1} \\
&= \left(1 + \frac{1 - p}{p} \cdot \frac{d\tilde{\mathbb{P}}_o(P_t^\pi)}{d\tilde{\mathbb{P}}_1(P_t^\pi)}\right)^{-1},
\end{aligned}
\tag{2}
$$

where the existence of the Radon-Nykodim derivative $\frac{d\tilde{\mathbb{P}}_o(P_t^\pi)}{d\tilde{\mathbb{P}}_1(P_t^\pi)}$ follows from the fact that $\tilde{\mathbb{P}}_o(P_t^\pi) << \tilde{\mathbb{P}}_1(P_t^\pi)$. Hence, we have that

$$\mu_t = \begin{cases} 1, & \text{for } t \geq \tau \\ \left(1 + \frac{1-p}{p} \cdot \frac{d\tilde{\mathbb{P}}_o(P_t^\pi)}{d\tilde{\mathbb{P}}_1(P_t^\pi)}\right)^{-1}, & \text{for } 0 \leq t < \tau. \end{cases}$$

Now we focus on evaluating $\frac{d\tilde{\mathbb{P}}_o(P_t^\pi)}{d\tilde{\mathbb{P}}_1(P_t^\pi)}$. Using a further application of Bayes' rule we have that

$$\left(\frac{d\tilde{\mathbb{P}}_o(P_t^\pi)}{d\tilde{\mathbb{P}}_1(P_t^\pi)}\right)^{-1} = \frac{d\mathbb{P}(\sigma(X(P_t^\pi)), t < \tau|\Theta = 1)}{d\mathbb{P}(\sigma(X(P_t^\pi)), t < \tau|\Theta = 0)}$$

$$= \frac{\mathbb{P}(t < \tau|X(P_t^\pi), \Theta = 1) \cdot d\mathbb{P}(X(P_t^\pi)|\Theta = 1)}{\mathbb{P}(t < \tau|X(P_t^\pi), \Theta = 0) \cdot d\mathbb{P}(X(P_t^\pi)|\Theta = 0)}$$

$$= \frac{d\mathbb{P}(X(P_t^\pi)|\Theta = 1)}{d\mathbb{P}(X(P_t^\pi)|\Theta = 0)} \cdot \mathbb{P}(t < \tau|X(P_t^\pi), \Theta = 1), \qquad (3)$$

where we have used the fact that $\mathbb{P}(t < \tau|X(P_t^\pi), \Theta = 0) = 1$. For any partition $P_t^\pi$, the *likelihood ratio* $\frac{d\mathbb{P}(X(P_t^\pi)|\Theta=1)}{d\mathbb{P}(X(P_t^\pi)|\Theta=0)}$ is an elementary predictable process that takes an initial value that is equal to the prior $p$ (when no samples are initially observed), and then takes constant values of $\frac{d\mathbb{P}(X(P_t^\pi)|\Theta=1)}{d\mathbb{P}(X(P_t^\pi)|\Theta=0)}$ in the interval between any two samples in the partition (only when a new sample is observed, the likelihood is updated). Hence, we have that

$$\frac{d\mathbb{P}(X(P_t^\pi)|\Theta = 1)}{d\mathbb{P}(X(P_t^\pi)|\Theta = 0)} = p\,\mathbf{1}_{\{t=0\}} + \sum_{k=1}^{N(P_t^\pi)-1} \frac{\mathbb{P}(X(P_t^\pi)|\Theta = 1)}{\mathbb{P}(X(P_t^\pi)|\Theta = 0)}\,\mathbf{1}_{\{P_t^\pi(k-1)\leq t\leq P_t^\pi(k)\}}.$$

The process is predictable since the likelihood remains constant as long as no new samples are observed. Modulated by the *survival probability*, $\left(\frac{d\tilde{\mathbb{P}}_o(P_t^\pi)}{d\tilde{\mathbb{P}}_1(P_t^\pi)}\right)^{-1}$ can be written as

$$p\,\mathbb{P}(\tau > t|\Theta = 1)\,\mathbf{1}_{\{t<P_t^\pi(k)\}} + \sum_{k=1}^{N(P_t^\pi)-1} \frac{\mathbb{P}(X(P_t^\pi)|\Theta = 1)}{\mathbb{P}(X(P_t^\pi)|\Theta = 0)}\,\mathbb{P}(\tau > t|\sigma(X(P_t^\pi)), \Theta = 1)\,\mathbf{1}_{\{P_t^\pi(k)\leq t\leq P_t^\pi(k+1)\}}.$$

Under usual regularity conditions on $\mathbb{P}(\tau > t|\sigma(X(P_t^\pi)), \Theta = 1)$ it is easy to see that $\left(\frac{d\tilde{\mathbb{P}}_o(P_t^\pi)}{d\tilde{\mathbb{P}}_1(P_t^\pi)}\right)^{-1}$ will have jumps only at the time instances in the partition $P_t^\pi$ and at the stopping time $\tau$, i.e. a total of $N(P_{T_\pi \wedge \tau}^\pi) + \mathbf{1}_{\{\tau<\infty\}}$ jumps at the time indexes in $P_{t\wedge\tau}^\pi \cup \{\tau\}$. $\qquad\square$

**Proof of Corollary 1**

Recall that from Theorem 1, we know that the posterior belief process can be written as

$$\mu_t = \mathbf{1}_{\{t\geq\tau\}} + \mathbf{1}_{\{t<\tau\}}\mathbb{P}(\Theta = 1|\tilde{\mathcal{F}}_t).$$

Hence, the expected posterior belief at time $t + \Delta t$ given the information in the filtration $\tilde{\mathcal{F}}_t$ can be written as

$$\mathbb{E}\left[\mu_{t+\Delta t}\,\Big|\,\tilde{\mathcal{F}}_t\right] = \mathbb{E}\left[\mathbf{1}_{\{t+\Delta t\geq\tau\}} + \mathbf{1}_{\{t+\Delta t<\tau\}}\mathbb{P}(\Theta = 1|\tilde{\mathcal{F}}_{t+\Delta t})\,\Big|\,\tilde{\mathcal{F}}_t\right]$$

$$= \mathbb{E}\left[\mathbf{1}_{\{t+\Delta t\geq\tau\}}\,\Big|\,\tilde{\mathcal{F}}_t\right] + \mathbb{E}\left[\mathbf{1}_{\{t+\Delta t<\tau\}}\mathbb{P}(\Theta = 1|\tilde{\mathcal{F}}_{t+\Delta t})\,\Big|\,\tilde{\mathcal{F}}_t\right]$$

$$= \mathbb{P}(\Theta = 1, t + \Delta t \geq \tau|\tilde{\mathcal{F}}_t) + \mathbb{P}(t + \Delta t < \tau|\tilde{\mathcal{F}}_t) \cdot \mathbb{E}\left[\mathbb{P}(\Theta = 1|\tilde{\mathcal{F}}_{t+\Delta t})\,\Big|\,\tilde{\mathcal{F}}_t \vee \{t + \Delta t < \tau\}\right], \qquad (1)$$

and hence $\mathbb{E}\left[\mu_{t+\Delta t}\,\Big|\,\tilde{\mathcal{F}}_t\right]$ can be written as

$$\mathbb{P}(t+\Delta t \geq \tau|\tilde{\mathcal{F}}_t, \Theta = 1)\cdot\mathbb{P}(\Theta = 1|\tilde{\mathcal{F}}_t) + \mathbb{P}(t+\Delta t < \tau|\tilde{\mathcal{F}}_t)\cdot\mathbb{E}\left[\mathbb{P}(\Theta = 1|\tilde{\mathcal{F}}_{t+\Delta t})\,\Big|\,\tilde{\mathcal{F}}_t \vee \{t + \Delta t < \tau\}\right],$$

which is equivalent to

$$\mathbb{E}\left[\mu_{t+\Delta t}\Big|\tilde{\mathcal{F}}_t\right] = (1 - S_t(\Delta t)) \cdot \mu_t + \mathbb{P}(t + \Delta t < \tau|\tilde{\mathcal{F}}_t) \cdot \mathbb{E}\left[\mathbb{P}(\Theta = 1|\tilde{\mathcal{F}}_{t+\Delta t})\Big|\tilde{\mathcal{F}}_t \vee \{t + \Delta t < \tau\}\right].$$
(2)

Furthermore, the term $\mathbb{P}(t + \Delta t < \tau|\tilde{\mathcal{F}}_t)$ in the expression above can be expressed as

$$\mathbb{P}(t + \Delta t < \tau|\tilde{\mathcal{F}}_t) = \mathbb{P}(t + \Delta t < \tau|\tilde{\mathcal{F}}_t, \Theta = 1) \cdot \mathbb{P}(\Theta = 1|\tilde{\mathcal{F}}_t) + \mathbb{P}(t + \Delta t < \tau|\tilde{\mathcal{F}}_t, \Theta = 0) \cdot \mathbb{P}(\Theta = 0|\tilde{\mathcal{F}}_t)$$
(3)

$$= S_t(\Delta t) \cdot \mu_t + (1 - \mu_t).$$

Therefore, $\mathbb{E}\left[\mu_{t+\Delta t}\Big|\tilde{\mathcal{F}}_t\right]$ can be written as

$$\mathbb{E}\left[\mu_{t+\Delta t}\Big|\tilde{\mathcal{F}}_t\right] = (1 - S_t(\Delta t)) \cdot \mu_t + (1 - \mu_t + S_t(\Delta t) \cdot \mu_t) \cdot \mathbb{E}\left[\mathbb{P}(\Theta = 1|\tilde{\mathcal{F}}_{t+\Delta t})\Big|\tilde{\mathcal{F}}_t \vee \{t + \Delta t < \tau\}\right].$$
(4)

Now it remains to evaluate the term $\mathbb{E}\left[\mathbb{P}(\Theta = 1|\tilde{\mathcal{F}}_{t+\Delta t})\Big|\tilde{\mathcal{F}}_t \vee \{t + \Delta t < \tau\}\right]$ in order to find $\mathbb{E}\left[\mu_{t+\Delta t}\Big|\tilde{\mathcal{F}}_t\right]$. We first note that

$$\mathbb{E}\left[\mathbb{P}(\Theta = 1|\tilde{\mathcal{F}}_{t+\Delta t})\Big|\tilde{\mathcal{F}}_t \vee \{t + \Delta t < \tau\}\right] = \mathbb{E}\left[\mathbb{P}(\Theta = 1|\sigma(X^\tau(P_{t+\Delta t}^\pi)), t + \Delta t < \tau)\Big|\tilde{\mathcal{F}}_t\right].$$

We start evaluating the above by first looking at the term $\mathbb{P}(\Theta = 1|\sigma(X^\tau(P_{t+\Delta t}^\pi)), t + \Delta t < \tau)$. Using Bayes' rule, we have that

$$\mathbb{P}(\Theta = 1|X^\tau(P_{t+\Delta t}^\pi), t + \Delta t < \tau) = \frac{\mathbb{P}(\Theta = 1, X^\tau(P_{t+\Delta t}^\pi), t + \Delta t < \tau)}{\mathbb{P}(X^\tau(P_{t+\Delta t}^\pi), t + \Delta t < \tau)},$$
(5)

where $\mathbb{P}(\Theta = 1, X^\tau(P_{t+\Delta t}^\pi), t + \Delta t < \tau)$ can be expanded using successive applications of Bayes' rule as

$$\mathbb{P}(\Theta = 1|X^\tau(P_t^\pi), t < \tau) \cdot \mathbb{P}(X^\tau(P_t^\pi), t < \tau) \cdot \mathbb{P}(t + \Delta t < \tau|\Theta = 1, X^\tau(P_t^\pi), t < \tau)$$
$$\cdot d\mathbb{P}(X^\tau(t + \Delta t)|\Theta = 1, X^\tau(P_t^\pi), t + \Delta t < \tau),$$

which is equivalent to

$$\mathbb{P}(\Theta = 1, X^\tau(P_{t+\Delta t}^\pi), t + \Delta t < \tau) = \mu_t \cdot S_t(\Delta t) \cdot \mathbb{P}(X^\tau(P_t^\pi), t < \tau) \cdot d\mathbb{P}(X^\tau(t + \Delta t)|\Theta = 1, X^\tau(P_t^\pi), t + \Delta t < \tau)$$
(6)

Similarly, it is easy to see that

$$\mathbb{P}(\Theta = 0, X^\tau(P_{t+\Delta t}^\pi), t + \Delta t < \tau) = (1 - \mu_t) \cdot \mathbb{P}(X^\tau(P_t^\pi), t < \tau) \cdot d\mathbb{P}(X^\tau(t + \Delta t)|\Theta = 0, X^\tau(P_t^\pi), t + \Delta t < \tau),$$
(7)

where again, we have used the fact that $\mathbb{P}(t + \Delta t < \tau|\Theta = 0, X^\tau(P_t^\pi), t < \tau) = 1$. Now we re-formulate (5) using Bayes rule to arrive at the following

$$\mathbb{P}(\Theta = 1|X^\tau(P_{t+\Delta t}^\pi), t + \Delta t < \tau) = \frac{\mathbb{P}(\Theta = 1, X^\tau(P_{t+\Delta t}^\pi), t + \Delta t < \tau)}{\sum_{\theta \in \{0,1\}} \mathbb{P}(\Theta = \theta, X^\tau(P_{t+\Delta t}^\pi), t + \Delta t < \tau)},$$
(8)

then using (6) and (7), (8) can be further reduced to $\mathbb{P}(\Theta = 1|X^\tau(P_{t+\Delta t}^\pi), t + \Delta t < \tau) =$

$$\frac{\mu_t \cdot S_t(\Delta t) \cdot d\mathbb{P}(X^\tau(t + \Delta t)|\Theta = 1, X^\tau(P_t^\pi), t + \Delta t < \tau)}{\mu_t \cdot S_t(\Delta t) \cdot d\mathbb{P}(X^\tau(t + \Delta t)|\Theta = 1, X^\tau(P_t^\pi), t + \Delta t < \tau) + (1 - \mu_t) \cdot d\mathbb{P}(X^\tau(t + \Delta t)|\Theta = 0, X^\tau(P_t^\pi), t + \Delta t < \tau)}.$$
(9)

Finally, we use the expression in (9) to evaluate the term $\mathbb{E}\left[\mathbb{P}(\Theta = 1|\sigma(X^\tau(P_{t+\Delta t}^\pi)), t + \Delta t < \tau)\Big|\tilde{\mathcal{F}}_t\right]$ as follows

$$\mathbb{E}\left[\mathbb{P}(\Theta = 1|\sigma(X^\tau(P_{t+\Delta t}^\pi)), t + \Delta t < \tau)\Big|\tilde{\mathcal{F}}_t\right] =$$

$$\sum_{\theta \in \{0,1\}} \int \mathbb{P}(\Theta = 1 | X^\tau(P^\pi_{t+\Delta t}), t + \Delta t < \tau) \cdot d\mathbb{P}(X^\tau(t + \Delta t) | \Theta = \theta, X^\tau(P^\pi_t), t + \Delta t < \tau),$$

which, using (9), can be written as

$$\sum_{\theta \in \{0,1\}} \int \frac{\mu_t \cdot S_t(\Delta t) \cdot d\mathbb{P}(X^\tau(t + \Delta t) | \Theta = 1, X^\tau(P^\pi_t), t + \Delta t < \tau) \cdot d\mathbb{P}(X^\tau(t + \Delta t) | \Theta = \theta, X^\tau(P^\pi_t), t + \Delta t < \tau)}{\mu_t \cdot S_t(\Delta t) \cdot d\mathbb{P}(X^\tau(t + \Delta t) | \Theta = 1, X^\tau(P^\pi_t), t + \Delta t < \tau) + (1 - \mu_t) \cdot d\mathbb{P}(X^\tau(t + \Delta t) | \Theta = 0, X^\tau(P^\pi_t), t + \Delta t}$$

Since

$$\sum_{\theta \in \{0,1\}} d\mathbb{P}(X^\tau(t + \Delta t) | \Theta = \theta, X^\tau(P^\pi_t), t + \Delta t < \tau) =$$

$$\mu_t \cdot S_t(\Delta t) \cdot d\mathbb{P}(X^\tau(t+\Delta t)|\Theta = 1, X^\tau(P^\pi_t), t+\Delta t < \tau) + (1-\mu_t) \cdot d\mathbb{P}(X^\tau(t+\Delta t)|\Theta = 0, X^\tau(P^\pi_t), t+\Delta t < \tau),$$

then the integral above reduces to

$$\int \mu_t \cdot S_t(\Delta t) \cdot d\mathbb{P}(X^\tau(t+\Delta t)|\Theta = \theta, X^\tau(P^\pi_t), t+\Delta t < \tau) = \mu_t \cdot S_t(\Delta t) \cdot \int d\mathbb{P}(X^\tau(t+\Delta t)|\Theta = \theta, X^\tau(P^\pi_t), t+\Delta t < \tau),$$

and since the conditional density integrates to 1, i.e. $\int d\mathbb{P}(X^\tau(t + \Delta t)|\Theta = \theta, X^\tau(P^\pi_t), t + \Delta t < \tau) = 1$, then we have that

$$\mathbb{E}\left[\mathbb{P}(\Theta = 1|\sigma(X^\tau(P^\pi_{t+\Delta t})), t + \Delta t < \tau)\Big|\tilde{\mathcal{F}}_t\right] = \mu_t \cdot S_t(\Delta t).$$

By substituting the above in (4), we arrive at

$$\mathbb{E}\left[\mu_{t+\Delta t}\Big|\tilde{\mathcal{F}}_t\right] = (1 - S_t(\Delta t)) \cdot \mu_t + (1 - \mu_t + S_t(\Delta t) \cdot \mu_t) \cdot \mathbb{E}\left[\mathbb{P}(\Theta = 1|\tilde{\mathcal{F}}_{t+\Delta t})\Big|\tilde{\mathcal{F}}_t \vee \{t + \Delta t < \tau\}\right]$$
$$= (1 - S_t(\Delta t)) \cdot \mu_t + (1 - \mu_t + S_t(\Delta t) \cdot \mu_t) \cdot \mu_t \cdot S_t(\Delta t)$$
$$= \mu_t - \mu_t^2 S_t(\Delta t)(1 - S_t(\Delta t)). \qquad (10)$$

Since $S_t(\Delta t) \geq 0, \forall t, \Delta t \in \mathbb{R}_+$, then the term $\mu_t^2 S_t(\Delta t)(1 - S_t(\Delta t)) \geq 0$, and it follows that

$$\mathbb{E}\left[\mu_{t+\Delta t}\Big|\tilde{\mathcal{F}}_t\right] \leq \mu_t, \forall t, \Delta t \in \mathbb{R}_+,$$

and hence the posterior belief process $(\mu_t)_{t \in \mathbb{R}_+}$ is a supermartingale with respect to the filtration $\tilde{\mathcal{F}}_t$.
□

**Proof of Theorem 2**

Assume a discrete-time version of the problem, where the decision $(\hat{\theta}^\pi_t, \delta^\pi_t)$ are made in time steps $\{0, \Delta t, 2\Delta t, \ldots\}$. Define a *value function* $V : \mathbb{N} \times [0, 1] \rightarrow \mathbb{R}_+$ as a map from the current history to the risk of the best policy given the history $\tilde{\mathcal{F}}_t$ as follows:

$$V(\tilde{\mathcal{F}}_t) \triangleq \inf_{(\hat{\theta}_\pi, T_\pi \geq t, P^\pi_{T_\pi} \supset P^\pi_t)} \mathbb{E}\left[\ell(\pi; \Theta)\Big|\tilde{\mathcal{F}}_t\right],$$

and define the *action-value function* as the value function achieved by taking actions $(\hat{\theta}_t, \delta_t)$, and then following the best policy thereafter. That is, when the decision is to *continue* (i.e. $\hat{\theta}_t = \emptyset$), we have that

$$Q(\tilde{\mathcal{F}}_t; (\hat{\theta}_t = \emptyset, \delta_t = 1)) \triangleq \inf_{(\hat{\theta}_\pi, T_\pi \geq t, P^\pi_{T_\pi} \supset P^\pi_t, t \in P^\pi_{T_\pi})} \mathbb{E}\left[\ell(\pi; \Theta)\Big|\tilde{\mathcal{F}}_t\right],$$

and

$$Q(\tilde{\mathcal{F}}_t; (\hat{\theta}_t = \emptyset, \delta_t = 0)) \triangleq \inf_{(\hat{\theta}_\pi, T_\pi \geq t, P^\pi_{T_\pi} \supset P^\pi_t, t \notin P^\pi_{T_\pi})} \mathbb{E}\left[\ell(\pi; \Theta)\Big|\tilde{\mathcal{F}}_t\right].$$

Based on Bellmans optimality principle [24], we know that the optimal policy has to satisfy the following in every time step, i.e.

$$\delta^{\pi^*}_t = \arg\inf_{\delta_t \in \{0,1\}} Q(\tilde{\mathcal{F}}_t; (\hat{\theta}_t = \emptyset, \delta_t)).$$

Now let us look at the optimal partition on $P^{\pi^*}_{T_{\pi^*}}$ on the discrete time steps $\{0, \Delta t, 2\Delta t, \ldots\}$, and look at an arbitrary realization for $P^{\pi^*}_{T_{\pi^*}}$. Then we pick two consecutive time indexes in $\{0, \Delta t, 2\Delta t, \ldots\}$,

say $n_1\Delta t$ and $n_2\Delta t$, with $n_1 < n_2$, for which $\delta_{n_1\Delta t}^{\pi^*} = \delta_{n_2\Delta t}^{\pi^*} = 1$, and $\delta_{n\Delta t}^{\pi^*} = 0, \forall n_1 < n < n_2$. Since the policy is optimal, we know that

$$\arg\inf_{\delta_n\Delta t \in \{0,1\}} Q(\tilde{\mathcal{F}}_{n\Delta t}; (\hat{\theta}_{n\Delta t} = \emptyset, \delta_{n\Delta t})) = 0, \forall n_1 < n < n_2,$$

and

$$\arg\inf_{\delta_{n_2\Delta t} \in \{0,1\}} Q(\tilde{\mathcal{F}}_{n_2\Delta t}; (\hat{\theta}_{n_2\Delta t} = \emptyset, \delta_{n_2\Delta t})) = 1,$$

which is equivalent to

$$\arg\inf_{\delta_n\Delta t \in \{0,1\}} \mathbb{E}\left[\ell(\pi; \Theta) \Big| \tilde{\mathcal{F}}_{n\Delta t}\right] = 0, \forall n_1 < n < n_2,$$

and

$$\arg\inf_{\delta_{n_2\Delta t} \in \{0,1\}} \mathbb{E}\left[\ell(\pi; \Theta) \Big| \tilde{\mathcal{F}}_{n_2\Delta t}\right] = 1,$$

which can be further decomposed into

$$\arg\inf_{\delta_n\Delta t \in \{0,1\}} \mathbb{E}\left[\ell(\pi; \Theta) \Big| \sigma(X(P_{n_1\Delta t}^{\pi^*})) \vee \mathcal{S}_{n\Delta t}\right] = 0, \forall n_1 < n < n_2,$$

and

$$\arg\inf_{\delta_{n_2\Delta t} \in \{0,1\}} \mathbb{E}\left[\ell(\pi; \Theta) \Big| \sigma(X(P_{n_1\Delta t}^{\pi^*})) \vee \mathcal{S}_{n_2\Delta t}\right] = 1,$$

since both functions $\mathbb{E}\left[\ell(\pi; \Theta) \Big| \sigma(X(P_{n_1\Delta t}^{\pi^*})) \vee \mathcal{S}_{n\Delta t}\right]$ and $\mathbb{E}\left[\ell(\pi; \Theta) \Big| \sigma(X(P_{n_1\Delta t}^{\pi^*})) \vee \mathcal{S}_{n_2\Delta t}\right]$ are $\tilde{\mathcal{F}}_{n_1\Delta t}$-measurable, then the decision-maker can compute the optimal decision sequence $\{\delta_{n\Delta t}\}_{n=n_1+1}^{n_2}$ at time $n_1\Delta t$. Since this holds for an arbitrary discretization step $\Delta t$, including an arbitrarily small step $\Delta t \to 0$, it follows that the sensing actions construct a predictable point process under the optimal policy, which concludes the Theorem. $\square$

**Proof of Theorem 3**

We start by proving that the optimal decision rule is $\mathbf{1}_{\left\{\mu_t > \frac{C_1}{C_o + C_1}\right\}}$. Fix an optimal stopping time $T_{\pi^*}$ and an optimal partition $P_{T_{\pi^*}}^{\pi^*}$. The optimal decision rule is given by

$$\hat{\theta}_{\pi^*} = \arg\inf_{\hat{\theta}_\pi} \mathbb{E}\left[\ell(\pi; \Theta) \Big| P_{T_{\pi^*}}^{\pi^*}, T_{\pi^*}\right],$$

which is equivalent to

$$\hat{\theta}_{\pi^*} = \arg\inf_{\hat{\theta}_\pi} \mathbb{E}\left[(C_1 \mathbf{1}_{\{\hat{\theta}_\pi=0,\theta=1\}} + C_o \mathbf{1}_{\{\hat{\theta}_\pi=1,\theta=0\}} + C_d T_{\pi^*}) \mathbf{1}_{\{T_{\pi^*}\leq\tau\}} + C_r \mathbf{1}_{\{T_{\pi^*}>\tau\}} + C_s N(P_{T_{\pi^*}\wedge\tau}^{\pi^*})\right],$$

which by smoothing can be written as

$$\hat{\theta}_{\pi^*} = \arg\inf_{\hat{\theta}_\pi} \mathbb{E}\left[\mathbb{E}\left[(C_1 \mathbf{1}_{\{\hat{\theta}_\pi=0,\theta=1\}} + C_o \mathbf{1}_{\{\hat{\theta}_\pi=1,\theta=0\}} + C_d T_{\pi^*}) \mathbf{1}_{\{T_{\pi^*}\leq\tau\}} + C_r \mathbf{1}_{\{T_{\pi^*}>\tau\}} + C_s N(P_{T_{\pi^*}\wedge\tau}^{\pi^*}) \Big| \tilde{\mathcal{F}}_{T_{\pi^*}}\right]\right],$$

and hence we have that

$$\hat{\theta}_{\pi^*} = \arg\inf_{\hat{\theta}_\pi} \mathbb{E}\left[\mathbb{E}\left[(C_1 \mathbf{1}_{\{\hat{\theta}_\pi=0,\theta=1\}} + C_o \mathbf{1}_{\{\hat{\theta}_\pi=1,\theta=0\}} + C_d T_{\pi^*}) \mathbf{1}_{\{T_{\pi^*}\leq\tau\}} \Big| \tilde{\mathcal{F}}_{T_{\pi^*}}\right] + \right.$$
$$\left. \mathbb{E}\left[C_r \mathbf{1}_{\{T_{\pi^*}>\tau\}} \Big| \tilde{\mathcal{F}}_{T_{\pi^*}}\right] + \mathbb{E}\left[C_s N(P_{T_{\pi^*}\wedge\tau}^{\pi^*}) \Big| \tilde{\mathcal{F}}_{T_{\pi^*}}\right]\right].$$

Since the terms $\mathbb{E}\left[C_r \mathbf{1}_{\{T_{\pi^*}>\tau\}} \Big| \tilde{\mathcal{F}}_{T_{\pi^*}}\right]$, $\mathbb{E}\left[C_d T_{\pi^*} \mathbf{1}_{\{T_{\pi^*}\leq\tau\}} \Big| \tilde{\mathcal{F}}_{T_{\pi^*}}\right]$, and $\mathbb{E}\left[C_s N(P_{T_{\pi^*}\wedge\tau}^{\pi^*}) \Big| \tilde{\mathcal{F}}_{T_{\pi^*}}\right]$ are the information and delay costs, which do not depend on the choice of $\hat{\theta}_\pi$, we have that

$$\hat{\theta}_{\pi^*} = \arg\inf_{\hat{\theta}_\pi} \mathbb{E}\left[\mathbb{E}\left[(C_1 \mathbf{1}_{\{\hat{\theta}_\pi=0,\theta=1\}} + C_o \mathbf{1}_{\{\hat{\theta}_\pi=1,\theta=0\}}) \mathbf{1}_{\{T_{\pi^*}\leq\tau\}} \Big| \tilde{\mathcal{F}}_{T_{\pi^*}}\right]\right],$$

which can be reduced to the following

$$\hat{\theta}_{\pi^*} = \arg\inf_{\hat{\theta}_\pi} \mathbb{E}\left[\mathbb{E}\left[(C_1 \mathbf{1}_{\{\hat{\theta}_\pi=0,\theta=1\}} + C_o \mathbf{1}_{\{\hat{\theta}_\pi=1,\theta=0\}}) \mathbf{1}_{\{T_{\pi^*}\leq\tau\}} \Big| \tilde{\mathcal{F}}_{T_{\pi^*}}\right]\right]$$
$$= \arg\inf_{\hat{\theta}_\pi} \mathbb{E}\left[C_1 \cdot \mathbb{E}\left[\mathbf{1}_{\{\hat{\theta}_\pi=0,\theta=1\}} \cdot \mathbf{1}_{\{T_{\pi^*}\leq\tau\}} \Big| \tilde{\mathcal{F}}_{T_{\pi^*}}\right] + C_o \cdot \mathbb{E}\left[\mathbf{1}_{\{\hat{\theta}_\pi=1,\theta=0\}} \cdot \mathbf{1}_{\{T_{\pi^*}\leq\tau\}} \Big| \tilde{\mathcal{F}}_{T_{\pi^*}}\right]\right]$$
$$= \arg\inf_{\hat{\theta}_\pi} \mathbb{E}\left[C_1 \cdot \mathbb{E}\left[\mathbf{1}_{\{\hat{\theta}_\pi=0\}} \cdot \mathbf{1}_{\{\theta=1\}} \cdot \mathbf{1}_{\{T_{\pi^*}\leq\tau\}} \Big| \tilde{\mathcal{F}}_{T_{\pi^*}}\right] + C_o \cdot \mathbb{E}\left[\mathbf{1}_{\{\hat{\theta}_\pi=1\}} \cdot \mathbf{1}_{\{\theta=0\}} \cdot \mathbf{1}_{\{T_{\pi^*}\leq\tau\}} \Big| \tilde{\mathcal{F}}_{T_{\pi^*}}\right]\right].$$
$$\tag{1}$$

Since $\mathbf{1}_{\{\hat{\theta}_\pi=\theta\}}$ is an $\tilde{\mathcal{F}}_{T_{\pi^*}}$-measurable function, we have that

$$\hat{\theta}_{\pi^*} = \arg\inf_{\hat{\theta}_\pi} \mathbb{E}\left[C_1 \cdot \mathbb{E}\left[\mathbf{1}_{\{\hat{\theta}_\pi=0\}} \cdot \mathbf{1}_{\{\theta=1\}} \cdot \mathbf{1}_{\{T_{\pi^*}\leq\tau\}} \Big| \tilde{\mathcal{F}}_{T_{\pi^*}}\right] + C_1 \cdot \mathbb{E}\left[\mathbf{1}_{\{\hat{\theta}_\pi=1\}} \cdot \mathbf{1}_{\{\theta=0\}} \cdot \mathbf{1}_{\{T_{\pi^*}\leq\tau\}} \Big| \tilde{\mathcal{F}}_{T_{\pi^*}}\right]\right]$$

$$= \arg\inf_{\hat{\theta}_\pi} \mathbb{E}\left[C_1 \cdot \mathbf{1}_{\{\hat{\theta}_\pi=0\}} \cdot \mathbb{E}\left[\mathbf{1}_{\{\theta=1\}} \cdot \mathbf{1}_{\{T_{\pi^*}\leq\tau\}} \Big| \tilde{\mathcal{F}}_{T_{\pi^*}}\right] + C_o \cdot \mathbf{1}_{\{\hat{\theta}_\pi=1\}} \cdot \mathbb{E}\left[\mathbf{1}_{\{\theta=0\}} \cdot \mathbf{1}_{\{T_{\pi^*}\leq\tau\}} \Big| \tilde{\mathcal{F}}_{T_{\pi^*}}\right]\right]$$

$$= \arg\inf_{\hat{\theta}_\pi} \mathbb{E}\left[C_1 \cdot \mathbf{1}_{\{\hat{\theta}_\pi=0\}} \cdot (1-\mu_{T_{\pi^*}}) + C_o \cdot \mathbf{1}_{\{\hat{\theta}_\pi=1\}} \cdot \mu_{T_{\pi^*}}\right]$$

$$= \arg\inf_{\hat{\theta}_\pi} \mathbb{E}\left[C_1 \cdot \mathbf{1}_{\{\hat{\theta}_\pi=0\}} \cdot (1-\mu_{T_{\pi^*}}) + C_o \cdot \mathbf{1}_{\{\hat{\theta}_\pi=1\}} \cdot \mu_{T_{\pi^*}}\right], \tag{2}$$

which is simply minimized by setting $\hat{\theta}_\pi = 1$ whenever $C_1(1-\mu_{T_{\pi^*}}) > C_o\mu_{T_{\pi^*}}$, hence we have that $\hat{\theta}_\pi = \mathbf{1}_{\{\}}$.

Now we resume by first defining the value and the action-value functions, and find the policy characteristics under Bellman optimality conditions.

Define a *value function* $V : \mathbb{N} \times [0,1] \to \mathbb{R}_+$ as a map from the current history to the risk of the best policy given the history $\tilde{\mathcal{F}}_t$ as follows:

$$V(\tilde{\mathcal{F}}_t) \triangleq \inf_{(\hat{\theta}_\pi, T_\pi \geq t, P^\pi_{T_\pi} \supset P^\pi_t)} \mathbb{E}\left[\ell(\pi;\Theta) \Big| \tilde{\mathcal{F}}_t\right],$$

and define the *action-value function* as the value function achieved by taking actions $(\hat{\theta}_t, \delta_t)$, and then following the best policy thereafter, i.e.

$$Q(\tilde{\mathcal{F}}_t; (\hat{\theta}_t, \delta_t)) \triangleq \inf_{(\hat{\theta}_\pi, T_\pi \geq t+\delta_t, P^\pi_{T_\pi} \supset P^\pi_t \cup \{t+\delta_t\})} \mathbb{E}\left[\ell(\pi;\Theta) \Big| \tilde{\mathcal{F}}_t\right].$$

Bellman optimality condition requires that at any time step $t$, we have

$$(\hat{\theta}_t^{\pi^*}, \delta_t^{\pi^*}) = \arg\inf_{(\hat{\theta}_t, \delta_t) \in \{0,1\} \times \mathbb{R}_+} Q(\tilde{\mathcal{F}}_t; (\hat{\theta}_t, \delta_t)).$$

Recall from the proof of Corollary 1 that the belief process follows the following dynamics

$$\mu_{t+\Delta t} =$$
$$\frac{\mu_t \cdot S_t(\Delta t) \cdot d\mathbb{P}(X^\tau(t+\Delta t)|\Theta=1, X^\tau(P^\pi_t), t+\Delta t < \tau)}{\mu_t \cdot S_t(\Delta t) \cdot d\mathbb{P}(X^\tau(t+\Delta t)|\Theta=1, X^\tau(P^\pi_t), t+\Delta t < \tau) + (1-\mu_t) \cdot d\mathbb{P}(X^\tau(t+\Delta t)|\Theta=0, X^\tau(P^\pi_t), t+\Delta t < \tau)},$$

which depends only on $\mu_t$ and the most recent sample realization in the partition $P^\pi_t$, which we denote as $\bar{X}^\tau(P^\pi_t)$. Hence, the tuple $(t, \mu_t, \bar{X}^\tau(P^\pi_t))$ is a Markov process since $X^\tau(t)$ is Markovian, and the belief process follows the above Markovian dynamics, and time is deterministic. Since the survival probability depends only on $\bar{X}^\tau(P^\pi_t)$, we can write the action-value function as

$$Q(\tilde{\mathcal{F}}_t; (\hat{\theta}_t, \delta_t)) \triangleq \inf_{(\hat{\theta}_\pi, T_\pi \geq t+\delta_t, P^\pi_{T_\pi} \supset P^\pi_t \cup \{t+\delta_t\})} \mathbb{E}\left[\ell(\pi;\Theta) \Big| \mu_t, \bar{X}^\tau(P^\pi_t)\right],$$

and consequently, the optimal actions at every time step $t$ following Bellman conditions are given by

$$(\hat{\theta}_t^{\pi^*}, \delta_t^{\pi^*}) = \arg\inf_{(\hat{\theta}_t, \delta_t) \in \{0,1\} \times \mathbb{R}_+} \inf_{(\hat{\theta}_\pi, T_\pi \geq t+\delta_t, P^\pi_{T_\pi} \supset P^\pi_t \cup \{t+\delta_t\})} \mathbb{E}\left[\ell(\pi;\Theta) \Big| \mu_t, \bar{X}^\tau(P^\pi_t)\right].$$

Hence, at any time step $t$, we only need to know the tuple $(t, \mu_t, \bar{X}^\tau(P^\pi_t))$ in order to compute the optimal action-value function, and hence, on the path to the optimal policy, knowing only $(t, \mu_t, \bar{X}^\tau(P^\pi_t))$ suffice to generate the random process $(T_{\pi^*}, P^{\pi^*}_{T_{\pi^*}}, \hat{\theta}_{\pi^*})$. Hence, $(t, \mu_t, \bar{X}^\tau(P^\pi_t))$ is a Markov sufficient statistic for $(T_{\pi^*}, P^{\pi^*}_{T_{\pi^*}}, \hat{\theta}_{\pi^*})$.

Note that our proof for the optimal decision rule $\hat{\theta}_{\pi^*}$ implies that the action-value function for stopping at time $t$, i.e. $\hat{\theta}_t^{\pi^*} \neq \emptyset$ is

$$Q(t, \mu_t, \bar{X}^\tau(P^\pi_t); (\hat{\theta}_t \neq \emptyset, \delta_t)) = C_o\mu_t \wedge C_1(1-\mu_t) + C_d t + C_s N(P^\pi_t),$$

whereas the continuation cost at any time step $t$ is given by finding the optimal rendezvous time $\inf_{\delta_t \in \mathbb{R}_+} Q(t, \mu_t, \bar{X}^\tau(P^\pi_t)); (\hat{\theta}_t = \emptyset, \delta_t))$. Therefore, the optimal action-value at any time step $t$ is given by

$$Q^*(t, \mu_t, \bar{X}^\tau(P^\pi_t); (\hat{\theta}_t \neq \emptyset, \delta_t)) = \min\{C_o\mu_t \wedge C_1(1-\mu_t) + C_d t + C_s N(P^\pi_t), \inf_{\delta_t \in \mathbb{R}_+} Q(t, \mu_t, \bar{X}^\tau(P^\pi_t); (\hat{\theta}_t = \emptyset, \delta_t))\}.$$

The equation above determines the stopping and continuation conditions, and using the monotonicity of the survival function in both time $t$ and the time series realizations $\bar{X}^\tau(P_t^\pi)$, we can show the monotonicity of the continuation set $\mathcal{C}(t, \bar{X}^\tau(P_t^\pi))$ using the same arguments of Theorem 1 in [15].

The optimal rendezvous can be found by optimizing the time interval such that the cost of stopping in the next time step is minimized. Hence, we have that

$$
\begin{aligned}
\delta_t^{\pi^*} &= \inf_{\delta_t \in \mathbb{R}_+} Q(t, \mu_t, \bar{X}^\tau(P_t^\pi); (\hat{\theta}_t = \emptyset, \delta_t)) \\
&= \inf_{\delta_t \in \mathbb{R}_+} \mathbb{E}\left[ \left(C_o \mu_{t+\delta_t} \wedge C_1(1 - \mu_{t+\delta_t}) + C_d t + \delta_t\right) \mathbf{1}_{\{t+\delta t < \tau\}} + C_r \mathbf{1}_{\{t+\delta t \geq \tau\}} + C_s N(P_t^\pi) + 1 \,\Big|\, \tilde{\mathcal{F}}_t \right] \\
&= \inf_{\delta_t \in \mathbb{R}_+} \left( (C_1 - C_o)\mathbb{P}(\mu_{t+\Delta t} \geq \frac{C_1}{C_o + C_1}) + C_1 \right) S_t(\delta_t) + C_r(1 - S_t(\delta_t)), \quad\quad (3)
\end{aligned}
$$

where $\mathbb{P}(\mu_{t+\Delta t} \geq \frac{C_1}{C_o+C_1})$ can be written as $\mathbb{P}(I_t(\Delta t) \geq \frac{C_1}{C_o+C_1} - \mu_t)$, where $I_t(\Delta t) = \mu_{t+\Delta t} - \mu_t$ is the information gain. $\qquad\square$