[Reviews · NeurIPS 2016]

Reviewer 1

Summary

The authors present a Bayesian model of decision making in which the observer chooses a policy about when to observe information in order to decide whether or not an adverse event is about to happen. The paper is well-written and clearly outlines theoretical results about the sampling policy. For instance, the observer can compute the next observation time after observing a sample.

Qualitative Assessment

My primary concern is the lack of practical applications in this paper. The formal framework given here is interesting on its own, but I'm not sure what someone in cognitive science working on 2AFC decision tasks (one of the motivations given in the introduction) can take from this. Minor: possible typo on page 8, line 300 "survival risk survival"?

Confidence in this Review

2-Confident (read it all; understood it all reasonably well)


Reviewer 2

Summary

The paper presents a continuous-time model of decision-making under time pressure. The decision-maker's goal is to anticipate whether a given bad event will occur. To do so, she can decide whether to halt and make a prediction, or perform discrete measurements of an auxiliary stochastic process. The setup and optimality criterion follows the standard subjective expected utility (i.e. Bayesian) setup. The main result of the paper is that optimal policies have a so-called "rendez-vouz" structure, which effectively transforms the continuous-time problem into a discrete-time problem (albeit the resulting information-MPD has a continuous state-space); furthermore, the optimal schedule for the next measurement is shown to be a trade-off between an "information gain" (=change of posterior) and a "suspense" function (=prob. of failure).

Qualitative Assessment

Pros: - The mathematical formalization of the problem is neat. I especially liked how the model automatically yields a distinction between passive and active information gathering; and how the continuous-time problem is reduced to the discrete-time case. The conceptualization of the time of the next measurement as a trade-off between an "information-gain" and a "suspense" term is also interesting. Cons: - The intended target audience is relatively small and must have a solid background in the measure-theoretic treatment of martingales to benefit from the presented results. - Although the identification of an "information-gain" term and a "suspense" term is interesting, I fail to see how to leverage this conceptual separation. Also: how should I compute those terms? Perhaps the authors would like to elaborate on this a bit more. - The most important shortcoming of the paper (which IMHO is fatal) is the lack of a concrete solution method. As I understand it, the missing piece is a way to calculate the continuation set of Theorem 3. Did I miss something here? If this is clarified, then it would definitively render the paper very interesting for the community. POST-REBUTTAL COMMENT ===================== Thank you for addressing my concerns.

Confidence in this Review

2-Confident (read it all; understood it all reasonably well)


Reviewer 3

Summary

This paper presents a detailed theoretical analysis of a temporal decision making problem under stochastic deadlines. The optimal policy is characterized in terms of the tension between two quantities: suspense and surprise.

Qualitative Assessment

First, let me state that this is not my area of expertise, and I did not understand some technical aspects of the theory. On the positive side, this paper is highly rigorous, and addresses a problem which to my knowledge has not been analyzed in this particular way before. It contributes to an active area of research on decision making under uncertainty. On the negative side, I felt that the technical contribution was not accompanied by corresponding insight into applied problems. How is this new theory useful? No demonstrations were provided. I also felt that some aspects of the exposition felt that they were presenting old ideas in a very obscure way. For example, it seemed like the suspense-surprise trade-off is essentially a manifestation of the speed-accuracy trade-off; I realize that these aren't technically equivalent, but it seems to me that every suspense-surprise trade-off will induce a particular speed-accuracy trade-off, and it wasn't clear to me what the added insight is from the suspense-surprise formulation. Finally, the authors do not provide any algorithmic proposals for actually computing the optimal policy; this is important if the ideas are to be applied in practical settings. I'm happy to be corrected if any of these criticisms reflect misunderstandings of the paper, since I'm not confident I grasped all the technical details. I think the paper could be improved by providing more expository material, explaining some of the more obscure terms (e.g., compensator process), and relegating some of the technical details to the supplement. This would make the work accessible to a wider audience characteristic of NIPS. Currently, I think the work is appropriate for a hardcore probability theory audience.

Confidence in this Review

1-Less confident (might not have understood significant parts)


Reviewer 4

Summary

This paper proposes an interesting model of decision making in a situation where the sensory observation depends on the occurrence of the adverse event, i.e., the deadline. The subject needs to infer the deadline, i.e., occurrence of adverse event, based on the history of sensory information. The model demonstrates how to compute the optimal “date” for gathering the next information by balancing the Bayesian surprise and suspense. The optimal policy, the rendezvous policy, can be obtained using Bellman optimality condition. The papers also shows that the continuation and stopping regions depend not only on subject’s belief, but also the context of the sensory information. This paper is interesting and technically sound, though a practical application of this model is not clear. I expect that this paper should be able to draw certain attention from a small group of researchers in this community. However, I personally think this paper is more suitable for publication in a journal like SICON.

Qualitative Assessment

Technical quality: This paper proposes an interesting model and derives three main theorems. The paper demonstrates when the decision maker acquires new information (continuation) and when makes a final prediction (stop) in a scenario where the deadline depends on the sensory information. It also shows that the subject's belief state is a supermartingle, and that the optimal policy has a "rendezvous" structure. The continuation and stopping region in this model depend both on the subject's belief state and "context" of the sensory observations. The paper is interesting and technically sound. Novelty: Most decision making models for 2AFC and stop signal task assume that the deadline is independent of the sensory information. The main reason is because those cognitive tasks are designed in a particular way such that the sensory information does not convey the information about the occurrence of the deadline. For example, in 2AFC, the subject needs to decide the moving direction of the stimulus within a fixed or random deadline, which does not include any information about the stimulus. This paper proposes an interesting model of decision making in a situation where the sensory observation depends on the occurrence of the adverse event, i.e., the deadline. The model is interesting and has certain novelty. Potential impact : Although the model is conceptually novel, the structure of cost function is similar to previous Bayesian decision models in 2AFC. The dependence between observation and deadline can also be included in previous Bayesian decision models. From computational point of view, the impact of this paper is limited. It is not clear how we can build a practical algorithm and run simulations based on the result of this paper. Again, I think this paper is more suitable for publication in a theoretical journal because the proof of the theorem is critical and should at least be sketched in the paper. Clarity and presentation: This paper is well written and easy to follow. The author makes effort in explaining each theorem beside the equations. I would recommend the following improvement: 1. Provide the expected cost function to be minimized to give readers a clearer idea about the dependency between the deadline and observation context. 2. Use different notations for the sigma-algebra generated by the process' survival and by the stopping event. 3. Check a few typos existing in this paper. ----------------------- I have read the rebuttal. Like I said in my previous review, I like the main idea of this paper which tries to solve an important problem in decision making. My main concern is still the same. It is not clear how we can build a practical algorithm and run simulations based on the result of this paper. The whole paper is written in the language of probability measure theory and makes it very hard to be understood by the majority of the audience in the NIPS community. I definitely appreciate the theoretical contribution of this paper. However, I highly recommend the author submitting this paper to a more theoretically oriented journal in control theory or probability theory.

Confidence in this Review

2-Confident (read it all; understood it all reasonably well)


Reviewer 5

Summary

This paper provided a novel Bayesian model for deciding when to sample a dynamic signal and when to make judgement of a hidden variable, under the pressure that a potential adverse event might occur in an unknown time. Previous works have not dealt with the case in which the time before the adverse event happening has dependency with the dynamic signal. Thus this model might provide implications to real-world applications such as clinical decisions.

Qualitative Assessment

The framework appears quite novel to me. From a neuroscience point of view, it also appears interesting to test human behavior against this model under risky situation. The Rendezvous property of the optimal policy also appears quite attractive.

Confidence in this Review

1-Less confident (might not have understood significant parts)


Reviewer 6

Summary

This paper proposed a model for decision making with endogenous information acquisition under time pressure. The model can help understanding the nature of optimal decision-making in settings where timely risk assessment and information gathering is essential.

Qualitative Assessment

The model is theoretically sound but lack of practical evaluation.

Confidence in this Review

1-Less confident (might not have understood significant parts)